# Improved Wireless Medical Cyber-Physical System (IWMCPS) Based on Machine Learning

**DOI:** 10.3390/healthcare11030384

**Published:** 2023-01-29

**Authors:** Ahmad Alzahrani, Mohammed Alshehri, Rayed AlGhamdi, Sunil Kumar Sharma

**Affiliations:** 1Department of Information Technology, Faculty of Computing and Information Technology, King Abdulaziz University, Jeddah 21589, Saudi Arabia; 2Department of Information Technology, College of Computer and Information Sciences, Majmaah University, Majmaah 11952, Saudi Arabia

**Keywords:** security schemes, machine learning, medical cyber-physical systems, attacks, data, classification

## Abstract

Medical cyber-physical systems (MCPS) represent a platform through which patient health data are acquired by emergent Internet of Things (IoT) sensors, preprocessed locally, and managed through improved machine intelligence algorithms. Wireless medical cyber-physical systems are extensively adopted in the daily practices of medicine, where vast amounts of data are sampled using wireless medical devices and sensors and passed to decision support systems (DSSs). With the development of physical systems incorporating cyber frameworks, cyber threats have far more acute effects, as they are reproduced in the physical environment. Patients’ personal information must be shielded against intrusions to preserve their privacy and confidentiality. Therefore, every bit of information stored in the database needs to be kept safe from intrusion attempts. The IWMCPS proposed in this work takes into account all relevant security concerns. This paper summarizes three years of fieldwork by presenting an IWMCPS framework consisting of several components and subsystems. The IWMCPS architecture is developed, as evidenced by a scenario including applications in the medical sector. Cyber-physical systems are essential to the healthcare sector, and life-critical and context-aware health data are vulnerable to information theft and cyber-okayattacks. Reliability, confidence, security, and transparency are some of the issues that must be addressed in the growing field of MCPS research. To overcome the abovementioned problems, we present an improved wireless medical cyber-physical system (IWMCPS) based on machine learning techniques. The heterogeneity of devices included in these systems (such as mobile devices and body sensor nodes) makes them prone to many attacks. This necessitates effective security solutions for these environments based on deep neural networks for attack detection and classification. The three core elements in the proposed IWMCPS are the communication and monitoring core, the computational and safety core, and the real-time planning and administration of resources. In this study, we evaluated our design with actual patient data against various security attacks, including data modification, denial of service (DoS), and data injection. The IWMCPS method is based on a patient-centric architecture that preserves the end-user’s smartphone device to control data exchange accessibility. The patient health data used in WMCPSs must be well protected and secure in order to overcome cyber-physical threats. Our experimental findings showed that our model attained a high detection accuracy of 92% and a lower computational time of 13 sec with fewer error analyses.

## 1. Introduction to Cyber-Physical Systems

Cyber-physical systems (CPSs) have drawn attention from academia, businesses, and the government due to their extensive social, economic, and environmental impacts [1]. The potential for CPSs to positively impact society, the economy, the environment, and individual citizens has resulted in a surge of interest in the field in recent years. More importantly, dominance and innovation in information transmission systems have emerged from the rapid advancements in computer, communication, and storage systems. Although there is no proper definition, the future generation of systems with integrated communications, computing, and management is mainly known as cyber-physical systems, which aim to achieve reliability, efficiency, and strength with respect to biological data. At the same time, studies conducted to achieve these aims focus mainly on achieving security inside the CPS [2,3].

Given the widespread integration of cyber-physical systems in many vital infrastructures, any violations of the safety of these networks might be catastrophic [4]. For example, if a communication system between vehicles is affected and false information about distance is conveyed, an accident can occur. Indeed, as customers rely on all automobile choices, the introduction of autonomous automobiles has considerably worsened the situation [5].

In addition to concerns about security, the confidentiality of CPSs is another major problem. Cyber-physical systems are frequently widely spread across vast geographical areas and generate enormous volumes of information for data analysis and decision making [6]. Data collection allows the system to make intelligent choices using advanced algorithms [7]. Furthermore, data infringement can also occur in any system element, encompassing the data collection stages, video streaming, processing, and backup [8]. Again, many current CPS design techniques do not consider data security, which jeopardizes any data obtained [9].

Extensive research has been conducted on the difficulties and obstacles associated with building such a CPS, offering an extensive range of systems and interdisciplinary studies. These challenges include the electricity grid, environmental protection, communications, and surveillance systems [10]. Some of these difficulties are related to the CPS itself, architectural and platform development, hardware and software layout, and applications and interactions with people (such as boundaries, abstractions, quality of service (QoS) needs, etc.) [11].

We will not describe these issues and obstacles in detail here; instead, two instances are given. First, CPSs should satisfy security and dependability using real-time requirements [12]. Since the external structure often differs from the network structure, minor adjustments should be noted and addressed promptly [13]. In addition, the necessity for compatibility and adaptability between various systems should also be addressed [14]. Second, CPSs are commonly utilized in medicine due to their high capacity for communications, computing, and safety. When protecting the confidentiality of identities, the base station is connected to a random variable and encrypted for anonymity in the face of specific attacks. Theoretically, if an attacker cannot access the data center or gateway, it is thought to act as a command center. Encrypted communication models can be developed to aid in the creation of hybrid keys and to prevent data collection or construction in the headings. Automated malware recognition and prevention for cyber-physical methods are crucial in healthcare for detecting and categorizing malicious threats. Machine learning techniques can be used to detect and classify different attacks concerning data from medical records. The data stored in the cloud should be confidential for future use. In addition, it is crucial to comprehend the external process channels, such as the responses of human operators to the system, which include the MCPS, medical sensors, and devices. High-confidence CPSs and MCPSs take into account the rising infrastructure problem to provide security and privacy. However, conventional methods of analysis are inadequate for both understanding their complexity and forecasting how these systems should behave. For instance, as an increasing amount of Internet of Things (IoT) medical equipment and sensors interact with the real world and people, the bar for security, reliability, privacy, and safety increases dramatically for MCPSs. Therefore, cutting-edge technology is necessary for engineering system designers to construct and hypothesize complex interdependencies. Where typical real-time performance is insufficient for geographically large CPSs, novel connections between the physical and cyber components require new architectural paradigms. The availability of communication channels, which need security, is typically not taken into account in the CPS or the MCPS since the feedback and input are from the physical world. In terms of the CPS, this is a very important distinguishing feature.

For instance, different biosensors can obtain data from individuals at home and interact with a more powerful computing-capable third-party cloud service. Meanwhile, healthcare doctors can immediately evaluate the patient’s physical state and provide recommendations or medications.

Distributed denial of service (DDoS) assaults pose a severe danger to the stability of the Internet. A distributed denial of service (DDoS) attack aims to overwhelm a centralized system with so many queries or communications that it crashes, disrupting service for legitimate customers [15].

The advent of 5G has opened a new door to non-contact automated healthcare surveillance, which can be facilitated by collecting and interpreting relevant vehicle locations using vehicular fog computing. One study presented a COVID-19 vehicle based on an effective joint identification system for 5G-enabled vehicle fog computing to contain the current outbreak [16].

In another study, a modular square root-based defense mechanism was presented to prevent DoS attacks in 5G-enabled car networks. When applied to vehicle networks, the MSR-DoS solution guaranteed source authenticity and message integrity, and pseudonym privacy could not be linked, was traceable, and could be revoked under the tenets of Burrows–Abadi–Needham (BAN) logic. The research and performance comparison revealed that the MSR-DoS system had lower communication and computational expenses than the latest current works [17].

To address threats and ensure the safety of patient data, CPSs with machine learning techniques can be implemented for monitoring healthcare networks with anomalies and malicious activity. Malware, encryption, data theft, hacking, and threat actors remain common in the healthcare industry. 

This research focuses on building a safe architecture for CPS implementation by bringing together sensors, communications, computing, and security cores. Resource management and security concerns, such as cloud services, real-time planning, etc., are studied [18]. Healthcare data is vulnerable to different cyber-attacks and threats. Problems related to security, reliability, and transparency are considered in our IWMCPS, where patient data are secured from attacks such as denial of service and data injection.

The contributions of this article are as follows:(1)A thorough CPS-related academic survey of communication and networking is presented;(2)We introduce an IWMCPS architecture consisting of three elements: (1) the communications and monitoring core; (2) the computational and safety core; and (3) the real-time planning and administration of resources;(3)A mathematical model is used to analyze the error and decision capacity of the proposed model.

The remaining paper is organized as follows: Section 2 discusses the background of medical cyber-physical systems [19]. The proposed improved wireless medical cyber-physical system (IWMCPS) is developed and illustrated in Section 3. Section 4 depicts the software analysis of the proposed model. Finally, Section 5 includes our conclusions and recommendations for the scope of future research [20].

## 2. Background of Medical Cyber-Physical Systems

### 2.1. Cyber-Physical Systems

Personal data encryption has been thoroughly researched as a viable means of solving problems with CPSs. It has attracted the attention of professionals and academics from numerous disciplines and led to various powerful techniques. For example, Raju et al. [21] presented a reversible dual encryption method for real-time, chaotic-based network information. 

Nair, M. M. et al. [22] reported on duplicate network data. In addition, the chaotic sequences encryption method was used for the first time to scratch the position of network data worldwide.

Meng, W. et al. [23] proposed that minor changes in the external structure should be reported and addressed as soon as possible because the network structure is often different from the exterior structure. In the second period, the 0 and 1 bits of the targeted pixel value were screwed again, depending on some other chaotic events, to protect the selected (dense) specific data from being attacked. A pseudo pixel was built into this area by inserting the data as a substitute for the target pixel and quickly including the data. 

Liu, L. et al. [24] reported that integrated information could be immediately retrieved from ciphertext based on the necessary computation. When the recipient decrypts the data, every bit of the targeted pixel is recovered to obtain the authentication tokens hidden beneath double protection.

Celdrán, A. H. et al. [25] validated these findings and demonstrated that the algorithms offered quick and practical input and high capacity. However, because some disseminated data were not considered, the method had difficulties with poor data recovery and high latency when querying data. Celdrán et al. presented a networking data protection digital data encryption method based on public critical symbolic frequency identification to address the issue of not having a high level of existing anti-deciphering capability.

Fu, Z. et al. [26] reported on the digital data encryption of a network security object. The encryption was conducted using homomorphic symbol resonance detection to increase anti-deciphering levels, with essential optimizations of digital cryptography. Eventually, an experiment with simulations was carried out. The delay findings demonstrated that strong digital encryption increases cryptography and efficiently controls the decoding rate of confidential information. However, the duration for querying information was excessive. 

Li et al. [27] presented a technique for encrypting security information in an anti-peep system. As per the authors’ professional expertise and the architecture of the data, the anti-peep information was chosen and the choice outcomes were preprocessed. A variable window was provided to establish and delete unnecessary information per the productive projects’ characteristics. 

Mowla, N. I. et al. [28] discussed a procedure where the variable frame’s lowest, maximal, and minimal amount thresholds were fixed before starting the window. Afterward, the corresponding data were set, and the 3G technique was used to segment the message.

Chng, C. B. et al. [29] discussed the outcome of the computation being recorded in two-dimensional arrays after computing the resemblance of the information to be compared to the Windows data file. Furthermore, data record resemblance was calculated for missing columns, and each database table was duplicated in the system. Finally, the duplicate data were cleaned up for the conclusion. The process treated plaintext as an ongoing bitstream sent to the private networks and entered its link in the data security.

Gatouillat, A. et al. [30] used a seed to initialize the critical generation and generating sequences and created a random number with the randomized encryption algorithm. Finally, the secured bitstream was encoded with the encoded number. Subsequent analyses showed that the anti-attack data component was more significant.

Keerthana, K. et al. [31] determined the fundamental limits of privacy in future CPSs, which are human-centric, and found an ideal trade-off between privacy and the precision of community reconstructions among data security and trustworthiness purposes. Finally, the authors presented the Tru-Alarm technique, which identifies reliable alerts and makes CPSs more feasible. 

### 2.2. QoS problems in CPSs

Fiaidhi et al. [32] examined many important QoS issues, including resource restrictions, platform diversity, dynamic system architecture, and mixed congestion, among based processes.

Verma, P. et al. [33] examined QoS via a simulation using a genuine link-layer architecture and suggested a solution that benefits from existing prediction methods. The simulation data were presented to assess the effectiveness of several prediction methods.

Fan, C. F. et al. [34] focused mainly on communication, connectivity, and sensor-related CPSs, and two additional designs were also studied. First, in the UIUC research laboratory, the investigators addressed the primary problems associated with integrating sensors, personal computers, and network systems with autonomous data centers, groups, networks, and the Internet. The structure comprised four units, including medical, hospitable, residential, and supervisory networks.

Kazakov, I. D. et al. [35] examined a cyber-physical information system and focused on identifying events, trustworthy data processing, space-time analysis, confidentiality, and cyber-physical data system safety. Their significant components included height assessment, sleep surveillance, and other data networks for body sensors. Unlike their previous work, this study focused on using and incorporating cloud computing technologies and intelligent universal healthcare to enhance the current system’s efficiency while considering cyber-physical principles. 

Vaziri, M. et al. [36] introduced a new method for secure color image encryption using a chaotic system decoded with a simple, one-round procedure. In the first step, a hash function creates a cryptographic key that will be utilized in both the confusion and distribution phases. The experimental analysis showed that this method was more secure than previous approaches and was immune to various attacks, including differential attacks and occlusion threats.

Testa, A. et al. [37] examined two practical techniques and offered a what-if analysis and a robustness assessment, enabling lead designers to implement ideal wireless sensor network options from the standpoint of connectivity and information provision resilience by employing a formalized method centered on the incident calculus formal languages.

Sangjun Kim et al. [38] reported the dangers faced by CPSs and the security measures taken to counter them based on machine learning. A CPS architecture was outlined, dividing its components and their respective functions into the hardware, network, and software. The cyber-physical attacks were divided into their respective taxonomies, focusing on those that leverage the physics of the system they aim to disrupt.

Nilufer Tuptuk et al. [39] reviewed the current status of cyber-security studies aimed at making our nation’s water and sewage systems safer for the general public, including the current state of cyber-security for water systems, the challenges that need to be overcome, and the number of publications in this field.

Qiyi He et al. [40] produced a new connected and autonomous vehicle (CAV) communication cyber-attack dataset (called CAV-KDD, where KDD is the dataset). Data were collected on cyber-attacks on CAVs in this center with respect to their reliance on communication. The CAV-KDD training dataset was used to train two classification models using the decision tree and naïve Bayes machine learning methods. These two models were evaluated and analyzed in terms of their accuracy, precision, and duration when detecting each variety of communication-based assaults. 

### 2.3. Medical Cyber-Physical Systems

Shu H. et al. [41] presented a two-layer system paradigm in which health records are kept outside the blockchain but can be shared there. A certificateless aggregate signature strategy was introduced for blockchain-based medical cyber-physical systems (MCPSs) relying on the suggested multi-trapdoor hash function. This method satisfied the stringent security standards required for MCPSs without sacrificing efficiency in either computing or storage.

Iosif Progoulakis et al. [42] illustrated the development of cyber-physical systems in the marine sector and their increasing cyber weaknesses, describing various cyber threats, hazards, and known security breach situations. The authors suggested that merging the various physical and cyber security responsibilities might further improve the interaction between stakeholders, enhance functional and technical cyber and physical security robustness, and raise practical cyber security knowledge in coastal facilities.

Maryam Bagheri et al. [43] focused on building trust in machine learning-based forecasts and creating a safety guarantee claim for machine learning controls in learning-enabled MCPSs. As an example of a learning-enabled MCPS implementation, the safety promotion case was discussed for artificial pancreas systems (APSs), which offer a deep neural network implementation for predicting insulin levels.

Integrating medical care from many perspectives increases the confidentiality and privacy of patients’ real-time databases. In addition, security and data quality must be considered during the configuration phase of any activity involving security against different attacks. Security issues are the major drawback of CPSs. Our proposed IWMCPS addresses these security gaps through strategic planning based on the cloud storage and safety components. 

The confidentiality and privacy of patient details need to be protected from different attacks. All data stored in the database must be well protected by implementing security measures against different attacks. These security issues are considered in the proposed IWMCPS. 

This article presents an IWMCPS structure with several components and subsystems, summing up our fieldwork over the previous three years. The design of each element, computer, and strategic planning core inside the IWMCPS were discussed in depth, including the cloud and safety components. A situation involving its implementation in medical services also showed that the IWMCPS design was improved.

## 3. Proposed Improved Wireless Medical Cyber-Physical System (IWMCPS)

There are four standard components in MCPS architectures: (i) the information acquisition level; (ii) the preprocessing information level; (iii) the cloud processing level; and (iv) the action level. The operational and safety criteria for each class are presented in this section.

### 3.1. Data Acquisition Level

A body area network (BAN) is an information acquisition level consisting of wireless sensors worn for particular medical uses, such as collecting blood pressure, pulse, and skin temperature, or for data management on the cost of medical accessibility. A BAN makes medical information more accessible to patients by gathering and sending data to a nearby computerized device, such as a cloudlet. In BANs, Bluetooth or ZigBee technologies are used in battery-operated sensing devices, whereas RFID-operated passive sensor arrays do not require batteries. Sensors worn by patients to monitor their vital signs, such as blood pressure and skin temperature, are typically part of a body area network (BAN), which is a type of medical network. 

### 3.2. Data Aggregating Layer

Because of BAN detectors’ limited power, clouds or concentration are needed for an intermediary device. Sensors communicate the information obtained through Bluetooth to a bridge server (functioning as a concentrate). A block in the most significant IoT-based architectural building component enables separately weak gadgets to have generally strong capabilities through the concentration of the input from each instrument and the transfer of aggregates to the clouds. A cloudlet collects data from more capable machines, such as smartphones. Usually, a cloudlet has specialized Internet connections and is constructed using a separate computer. Generally, cloud-based architecture blocks enable weaker devices to perform at the same level as more powerful ones by converging data from each instrument and transferring them to the cloud, where they can be analyzed.

### 3.3. Storing and Cloud Computing Level

Since proper diagnosis requires long-term individual health surveillance, the cloud’s primary purpose is safe storage. Furthermore, public health rules require medical history to be stored for an extended period. Many cloud providers save health data by entering business association agreements (BAAs). Healthcare centers manage apps in their cloud environment (i.e., data centers) and consequently use the cloud for a second significant reason: computing. Therefore, confidentiality on a cloud platform is possible only with sophisticated cryptography systems. The third task of the cloud is data analysis, which helps medical practitioners make decisions by applying statistically inferred algorithms to the information they have gathered and forecasting patient health status. In recent years, these techniques have attracted attention for use in remote health surveillance systems.

Figure 1 shows the proposed IWMCPS model architecture. It has four levels: the acquisition, preprocessing, cloud, and activity levels. Initially, the data from the sensors are collected using the acquisition layer and then preprocessed in the next layer. The cloud layer stores and analyzes the processed data. The action layer visualizes the data to help improve decisions on the patient’s medical conditions. 

### 3.4. Action Level

The action level can be “dynamic” or “silent.” Proactive action uses an actuator, such as a mechanical system, to carry out an activity by utilizing the findings of algorithms that operate in the cloud. Robot-assisted surgeries are one example of this sort of action. No physical measures are taken in passive activities. The authority supports the results of the analysis of the findings of medical applications. The long-term (24 h) surveillance of a patient’s electrocardiogram (ECG) is an example of a simple activity that enables doctors to visualize the surveillance outcomes of 25–30 individuals within 10–20 min. First, data are gathered from the sensors using the acquisition layer and then preprocessed in the next layer. Finally, the processed information is stored and analyzed in the cloud. As shown in Figure 1, the action layer aids in analyzing patient medical conditions by providing a visual representation of the data.

### 3.5. IWMCPS Architectural Research

The logical structure of the IWMCPS divides communication, sensors, the computer core, etc., into smaller components. The data collected are either information on public health or information used to identify human actions in medical services. The buildings in the individual’s environment are linked to the detectors. The video technique is based on cameras that capture the scenery to acquire the moving item and infer actions such as walking, reclining, resting, leaning, jumping, leaping, running, etc. This represents a video-centered method. Authorization based on images and a monitoring method based on activities can improve safety and the versatility of user access. It should be noted that the safety core is not a distinct module used for communications and computing modules. For financial reasons, cloud processing services can also be found in the computer core. Each element or unit is discussed in detail in the following few sub-sections. Large-scale wireless communication differs from its typical ad hoc wireless counterparts. It uses fewer gadgets, operates at a higher level, and is more closely connected to its physical surroundings. WSNs require a limited power supply, which must be maintained for more extended use. However, various calculation and communication activities must be completed within time constraints to avoid unpleasant or disastrous consequences. Finding solid support for significant wireless sensor nodes is an important and challenging research problem.

#### 3.5.1. Core of Communications and Sensors

Innovations that merge computers and interfaces with the physical environment generate embedded systems and information networks. As mentioned above, wireless communication differs in that it operates on a larger scale, at a relatively high level, and with fewer gadgets and closer connections to the physical surroundings than its typical ad hoc wireless equivalents. Thus, the primary need is to enable a restricted power supply for wireless sensor networks (WSNs) to be maintained for a longer lifespan. In the meantime, various calculation and communications activities must be completed within temporal limits to avoid unpleasant or even disastrous results, given the essential nature of CPS systems. An important and challenging research problem is to provide solid help in the significant wireless sensor nodes. The security and adaptability of user access can be enhanced by using image-based authorization and activity-based monitoring. There is no separate module for the safety core; instead, it serves as a common platform for all communication and computing modules. For financial reasons, cloud computing services can also be seen in the computer core. The following sections provide more detail about each of the various components. 

The workflow of the proposed IWMCPS model is illustrated in Figure 2. This flow consists of sensor nodes, wireless sensors, wireless body area networks (WBANs), storage and actuation security models, cloud computing models, and threshold controllers. It illustrates the flow of data from sensed data to gateway data, which is saved for further analysis in several databases. Some key research concerns include planning, route, location, targeting medium accessibility, etc. Hierarchical communications can split the network into several jobs by reconfiguring mappings and processing methods. This ensures that the flexible and timely surveillance of MCPS patients can be effectively improved through the communications core. In a sensor network, the transmission is hard instantaneously but softer in actual environments. The module aims to construct an abstracted level in real time that requires newly distributed and real-time effective communications in wireless MCPSs with mobility elements under variable communication networks.

#### 3.5.2. Computation Core

Secure cloud computing services provide users of the higher layer, such as the community teams in hospitals, with programs for individual monitoring systems, atmospheric data processing, city traffic predictions, and analysis. Cloud computing is crucial in delivering high-performance computation, which is achieved by integrating smartphones in the cloud, enabling the use of many operating systems (OS), etc. It depicts the computational cloud architecture as a critical component in the center, including software as a service (SaaS), platform as a service (PaaS), and infrastructure as a service (IaaS). Two individual components, termed cloud providers and devices, are present in the cloud computing paradigm. Meanwhile, cloud computation can allow for the instantaneous delivery of services and applications. For further information, regular patients can consult the website of the lab. As a result, the communications core can provide more effective and timely monitoring of MCPS patients. Transmission in a sensor network can be hard or soft at any time, depending on the actual environment. As a result, new distributed and real-time effective wireless MCPS communications with mobility elements are required as a part of the module’s goal to construct an abstracted level in real time. For example, the cloud computing architecture has an entire server containing visual data related to traffic control. The traffic in a city can be predicted and analyzed by monitoring the visual data collected. The critical component of the cloud architecture is based on SaaS, PaaS, and IaaS, as illustrated in Figure 3. 

#### 3.5.3. Planning and Management of Resources in Real Time

Here, the conventional server center method is changed to the networking or the center of the cloud. Server farms have become server-virtualized systems that support hardware-aided simulations with innovative techniques. Innovative real-time scheduling and resource control techniques, which are economically customizable, are increasingly needed to fulfill users’ needs. A three-layered real-time planning and monitoring reference method is provided, which works with the communications and computing model.

End-to-end efficiency measures on the bottom level can be performed based on real-time data gathered by various sensors, cameras, and the server/client cloud. Some of these measures are related to managing the virtualized servers, creating virtual servers with the needed software, working on virtual machines (VMs), improving calculation time, reducing communication costs, and fulfilling deadlines and QoS requirements. In the middle level, priority is given to emergency reactions, such as healthcare, transport accidents, energy blackouts or power outages, etc. This involves the management and execution (upper layer) of assets (e.g., network connectivity, energy usage, computer resources (i.e., CPUs and memory use), and other support from servers and customers. It also includes the surveillance and tracking of supplies in real time.

#### 3.5.4. Security Core

The protection architecture of the proposed IWMCPS consists of two components: sensor and communication and storage and actuation protection. A decentralized mobile computer check mechanism is used as an efficient defense to cope with sensor and communications safety after coding the accurate sensed data using a shared or symmetrical key. A reduction is named if misconduct is reported to the control system, avoiding damage or assault to the opposite side of the network.

Concerning the confidentiality of identities, a connection between the base station and the random variable is used to ensure protection against assaults; in addition, the base station is used for anonymity in specific attacks. The gateways or the data center are supposed to serve as a central commanding agency that an adversary cannot breach. A secure communication model, which can assist in creating a hybrid key (asymmetric passcode and symmetrical keyed hash algorithm), can be developed to prevent data gathering or construction in the heading. A decentralized mobile computer check mechanism is an effective defense when a shared or symmetrical key protects sensors and communications. Misconduct can be classified as reduced if it is reported to the control system, which prevents damage or assault to the opposing network.

The only way to approach this is to disregard the sensor network identification (lack of privacy) asymmetrical key (between the sensor network and the ground station) while using asymmetric token haze to secure the detected data. The system is likewise impervious to the breach of the second version of the nodes, where the hazardous user enters the system. In this event, a failed check of the secrecy system will enable the ground station to identify this assault. Due to the inviolable computing features of the hashing algorithm, it is not simple for an opponent to corrupt the sensory node. Therefore, it is challenging for an opponent to find the keys to decode or access the communication. This also provides the node with an essential barrier when the key across uncommitted nodes becomes secret.

File restoration and failure recovery-based techniques identify illegal data alteration and destruction due to server breaches or random failures to address storage and actuating security. Upon processing, the data may have to be kept for subsequent transmission over time. Any misconduct or alteration of the stored data might lead to system-wide problems. As prediction accuracy increases, duplication is employed using the erasure coding approach to tolerate other malfunctions or server crashes in the cloud. Here, a set of criteria was implemented to validate an actuation procedure to guarantee no activation in an aggressive state without appropriate authorization.

### 3.6. Predicting and Analyzing Dependability

With the dependability of each part of the system, the overall system’s dependability can be calculated. The reliability of the system is likely since its desired output over a certain amount of time is generated by a device understanding that it operates flawlessly. Reliability has a value between 0 and 1, where 1 indicates 100% reliability. These reliability patterns represent a combined communication model for a system functioning with a conceptual order.

A system’s elements function separately inside it. Therefore, every aspect must operate appropriately so the system can perform well. This means that the original equipment must have a dependable setup. A single module’s failure leads the entire system to provide an incorrect output. The probabilities of the performance of the$\;Pr\left(in_{x}\right)$ components are defined by their dependability and are independent of the remainder of the network. A system comprises various parts that work independently of each other. As a result, each component must perform its function for the system to function correctly. In other words, the original equipment has a setup that ensures dependability. The entire system is affected by the failure of a single module.

The confidence in the overall system $S_{w}$ is the cross-section of the chance of occurrence. Because the fault rate of every element is continuous and irrespective of its use, the dependability of the xandm components equals $S_{w}$, or the number of components per work in a system. One element might be a key element throughout the calculations, as a module for the actuation or the system’s solitary actuator. The collected dataset is denoted in Equation (1).
(1)$$S=Pr_{w}=Pr\left\{in_{0},in_{1},\cdots ,in_{m}\right\}$$

In the above equation, probability is denoted as Pr; the input data is denoted as $in_{0},in_{1},\cdots ,in_{m}$; and the overall probability is denoted as $Pr_{w}$ and expressed in Equation (2):(2)$$Pr_{w}=\mathrm{Pr}\left(in_{0}\right)\times \mathrm{Pr}\left(in_{1}\right)\times \cdots \times \mathrm{Pr}\left(in_{m}\right)$$
where the overall probability of the network is denoted $Pr_{w}$ and the individual probability of each input is denoted as $\mathrm{Pr}\left(in_{0}\right),\mathrm{Pr}\left(in_{1}\right),\cdots ,\mathrm{Pr}\left(in_{m}\right)$. The chance of occurrence of the event is denoted in Equation (3):(3)$$S_{w}=\frac{1}{m}\sum_{x=0}^{m}S_{x}$$
where the system function occurrence is denoted as $S_{x}$ and the total number of available datapoints is denoted as m. The use of redundant elements impacts the system’s dependability. A redundancy device is an element that performs the same job as another element in the same category. For example, the duplicate element might be used in case of failure.

A pictorial representation of $S_{w}\;$is shown in Figure 3; it uses the probability of the input occurrence and the total number of input samples. All the input occurrences are added to find the overall event of the input. A network with dependencies can be demonstrated as reliable using Equation (4), where m represents the considerations when executing the same task.
(4)$$S_{w}=1-\frac{1}{m}\overset{m}{\underset{x=0}{\sum }}1-S_{x}$$

In Equation (4), the dependability of the single input is denoted as $S_{x}$ and the total number of datapoints available is denoted as m. The dependability is a fault rate to the time provided to determine the element’s reliability. The module efficiency is represented by Equation (5):(5)$$S={exp}^{-\beta t}+S_{w}$$
where S is the module’s efficiency, β is the element’s error rate, and t is the period. The overall system dependability is denoted as $S_{w}$. This metric concentrates on the system’s components (sensor, actuator, communications unit, controller, and computers) and modules for sensing (temperature probe, pressure transmitter, and gyroscopic detectors, among others). Maximum bandwidth, sensitivity, specimen frequency, distortion, hysteresis levels, measured error, precision, and failure rate are the major factors used to assess detection capability. The error rate is denoted in Equation (6).
(6)$$\beta_{T}=\frac{exp^{-\beta_{T}t}}{S_{w}}$$

Camera sensor dependability depends on its error rate ($\beta_{T}$). In Equation (6), the period is denoted as t and the overall system dependability is denoted as $S_{w}$. For a single action, specific systems may require more sensors. If s represents all the different types of sensors, m is the minimum sensory number necessary for the job, and $r_{C_{m}}$ is the number of ways that m can be chosen from r. Dependability can be appropriately developed after the new regime. 

For the actuator module, actor power, horsepower, velocity, working circumstances, and endurance are the standard evaluation criteria. The standard deviation is denoted in Equation (7):(7)$$S_{d}=\frac{exp^{-\beta_{d}t}}{1+\beta_{T}}$$
where the actuator’s dependability is expressed as $S_{d}$, $\beta_{d}$ is the equivalent fault rate, and the period is denoted as t. 

With respect to the communications module, the communications system brings data to the Internet from a physical environment. Network doorways serve as communication nodes and act as web servers. They are composed of interpreters, faults, and signal converters and employ various standards to ensure compatibility.

For the computer and controlling module, the mixture of operating systems is the computer and controlling unit. The Weibull dispersion and stochastic processes indicate the reliability of equipment and software. The programming is 100% trustworthy in embedded devices if the equipment provides correct input data. Furthermore, the environment is uncertain in complicated processes, such as for the targeted CPS, which means there is no guarantee of complete dependability. The technology is both time-independent and time-dependent.

The equipment and elements of programming are very interactive in CPSs. Thus, the various failure types can be categorized as (a) software configuration failure, (b) domain-specific inability, and (c) interface failure between development tools. The Weibull dispersion is a versatile function used to determine distributions according to scale, form, and location criteria. Therefore, equipment failures might best be represented using this model. In real-time situations, the Weibull model is a dependable and simple technique. The trustworthiness of technology is denoted in Equation (8):(8)$$S_{dev}=\frac{exp^{-{\left(\beta t\right)}^{\gamma }}}{1+\beta_{T}}$$
where β is the Gompertz distribution’s form element, γ is the aggregate failure rate of different hardware elements, the timing function is denoted as t, and the error rate is denoted as $\beta_{T}$. The programming system’s dependability is defined in Equation (9):(9)$$S_{soft}=\frac{exp^{m\left(t+i\right)-m\left(t\right)}}{1-\beta_{T}}$$
where m(t) is the accumulated time t error number and the error rate is denoted as $\beta_{T}$. Software–hardware interface problems are defined as software defects due to equipment configuration changes. These errors lead to changes in software performance across several profiles, and software errors often occur. Due to these shortcomings, development tools are incorporated into reliability modeling. Weibull distributed models, as opposed to normally distributed functions, depict software-interaction failures as equipment failure tolerances that occur during the equipment’s lifespan.

### 3.7. Medical Health Decisions

Here, we illustrate the application situation for medical treatment, which includes three primary venues: home, hospitals, and the workplace. This program was created to enhance care for older individuals whose children work in the workplace and conserve medical expenses. Typically, the subject will collect (via webcam, biomarker, mica, etc.) and store everyday physical and medicinal data in a third-party cloud (such as the central laboratory) using an in-home WSN cloud bridge. In-house physicians (or caregivers) can review this health file periodically and provide advice and medications via wired or wireless cloud connectivity. In an emergency scenario (such as an older person falling), information about this emergency is promptly provided to physicians and relatives to aid the individual.

The data processing module of the proposed IWMCPS model is shown in Figure 4. It receives data from the sensors and actuators. The communication network is used to store and process the data. The computation and control unit then use the hardware and software module to analyze the collected data. Wired and wireless connections are established between various systems. The clouds and the gateway allow for the calculation core (or even gadgets) to be constructed. Indoors, with different security techniques, real-time planning and resource monitoring are performed. It should be noted that various application solutions are built on the hospital side to improve health surveillance and available treatments. RFID tags or ZigBee gadgets such as MicaZare are used in areas where the use of cameras is not suitable. 

Cyber-physical systems are essential in the healthcare sector, which is vulnerable to information theft and cyber-attacks due to the life-critical and context-aware nature of health data. Reliability, confidence, security, and transparency are some of the issues that must be addressed in the growing field of MCPS study. To overcome the abovementioned problems, we proposed an improved wireless medical cyber-physical system (IWMCPS) based on machine learning techniques. The heterogeneity of devices included in health systems (such as mobile devices and body sensor nodes) introduces large attack surfaces. This necessitates effective security solutions for these environments based on deep neural networks for attack detection and classification. The large attack surfaces associated with all endpoint devices—from desktops to mobile phones, to USB ports, to carelessly disposed of hard drives—are all a part of the cyber-physical attack surface. In this study, we evaluated our design with actual patient data to determine its efficacy against security attacks, such as data modification, denial of service, and data injection. The simplest and most damaging form of network attack is a denial of service (DoS). It is utilized to achieve security control by requiring the system to ignore standard requests and send and receive data using the communication channel’s network resources. CPSs are essential to overcoming cyber-attacks. The body sensor nodes used in the proposed method are vulnerable to attacks. Therefore, effective security solutions based on deep neural networks were implemented for attack detection and classification. The evaluation of the proposed method was based on data modification and injection. 

In this way, the proposed IWMCPS was designed with WBAN, and the logical architecture of the proposed model was shown. The dependability and error of the proposed IWMCPS model were verified theoretically and using a mathematical model. A better decision model was used for improved results. 

## 4. Software Analysis

An effectiveness assessment computes the communications lag of a proposed system. It assesses the suggested prediction models in terms of effectiveness, half-total error rates, and computer effectiveness. With a 1.4 GHz 6x ARM Cortex, 1.8 GB RAM, and 64 GB of internal storage, we developed an onboard testbed utilizing inexpensive Raspberry Pi3 machines. A cloud atmosphere was provided that connected over 802.11 b networking with Raspberry Pi3 gadgets. In Raspberry Pi3 and the clouds, the data for the acquisition level were supplied. We utilized a computer with 64-bit Intel i9-58900 CPU 3.80 GHz and 16.00 GB RAM for cloud-assisted research. Cloud facilities were available when required. Three standard statistics were calculated to assess the prediction models. The model parameters were analyzed based on the internal structure and validated based on the data automatically, whereas the hyperparameters were set manually; the lambda architecture, which is based on the Apache Kafka and Apache Spark tools used for the simulation, was implemented in this system. 

Communication lag time is considered first when evaluating a system’s efficacy. The proposed prediction models were assessed for their effectiveness, half-total error rates, and computer effectiveness. Finally, the proposed model was evaluated in terms of accuracy, efficiency, delay, and time consumption. The data size variations are shown in the delay graph.

Figure 5a,b shows the results of the delay analysis of the existing CPS model and the proposed IWMCPS model. The simulation was performed by varying the data size from a minimum of 500 kB to a maximum of 5000 kB with a step size of 500 kB. The respective delay in producing the final output was measured for the proposed IWMCPS and compared with the existing model. The proposed IWMCPS model, with the better decision making and error dependability, produced a lower delay than the current model for all the situations. 

Table 1 shows the outcomes of the simulation analysis of the proposed IWMCPS. The outcomes, such as the accuracy and efficiency of the proposed IWMCPS, were analyzed and compared with existing models, including convolutional neural network (CNN), deep neural networks (DNN), random forest (RF), decision tree (DT), and fuzzy logic (FL). The accuracy and efficiency of the proposed IWMCPS were greater than those values observed for CNN, DNN, RF, DT, and FL. For particular data sizes, the accuracy and efficiency of the proposed method were higher. The results indicate that the proposed IWMCPS model has higher performance than the existing models. The proposed IWMCPS model was also compared with CNN and DNN based on classification accuracy and efficiency. The proposed IWMCPS model with its theoretical error calculation and avoidance model produced improved accuracy results. 

The results of the computation time analysis of the existing CPS model and the proposed IWMCPS model are depicted in Figure 6a,b. The simulation was carried out by varying the data size from the lowest to the highest value. Then, the respective time required to compute the final output was calculated and plotted. The proposed IWMCPS model required less time due to its minimal components and processing blocks. In contrast, the existing model required several layers and feedback models to produce optimized results. 

Table 2 shows the results of the computation time analysis for the proposed IWMCPS model. The computation time required to produce the final output was calculated for the proposed IWMCPS model and compared with the existing CPS model. As the data size increased, the computation time also increased. The proposed IWMCPS model had a shorter computation time than the existing CPS model due to its less complex processing layers. The proposed IWMCPS model, with its theoretical error calculation and avoidance model, produced faster accuracy results. 

Figure 7a,b shows the accuracy and efficiency analyses of the proposed IWMCPS model, respectively. The simulation output for the accuracy and efficiency of the proposed IWMCPS model was monitored and compared with existing models, including CNN, DNN, RF, DT, and FL. The results indicated that the proposed IWMCPS had higher performance due to its very low complexity and higher error prediction model. In addition, the complexity was reduced by reducing the number of processing layers. 

The effective protection of the environments was focused on deep neural networks for attack detection. The effective security solution was based on the attack detection rate, as shown in Figure 8. The attack detection rate was compared with CNN and DNN.

The nature of health data necessitates effective security solutions for these environments based on deep neural networks for attack detection and classification. In this study, we evaluated our design with actual patient data to assess its effectiveness against security attacks, including data modification, denial of service, and data injection.

The proposed IWMCPS model was designed, implemented, and tested. Simulation outcomes, such as accuracy, computation time, etc., were calculated for the proposed IWMCPS model and compared with the existing models. The results shown in the Figure 9. that the proposed IWMCPS model had higher performance due to its lower complexity and higher decision modules. According to the proposed model, traditional teaching methods can reduce latency decisions without relying on the cloud. Furthermore, there is evidence that the three bio-modalities evaluated in this article can provide good precision and computational complexity. In the future, we will investigate the use of compact deep neural networks for fast bio-modality classification. 

The evaluation of our proposed IWMCPS was based on delay analysis, simulation outcome analysis, computation time analysis, and efficiency analysis. The proposed model was compared with other existing methods and demonstrated the highest efficiency. Furthermore, the proposed method was compared with other machine learning techniques, including CNN and DNN. The overall performance of the system was determined; the attack detection rate was 98.92%, computation time was 13 sec, accuracy was 92%, and efficiency was 89%. 

## 5. Conclusion and Future Directions

Cyber-physical systems play a significant role in the design of future engineering technologies with a higher capacity than current counterparts. However, CPSs are still associated with several problems and obstacles. Here, we presented a new architecture incorporating sensing, communications, processing, cloud services, and safety cores. The structure also includes other components, such as security and cloud models. A physical scenario for the suggested improved wireless medical cyber-physical system (IWMCPS) structure was also described. The proposed traditional teaching model effectively reduces latency decisions without using the cloud. It was also shown that the three bio-modalities assessed in this article may deliver good precision and computation complexity. We plan to investigate the application of our model in faked classification tasks of bio-modalities using compact deep neural networks for speed performance. However, the data size is currently limited to a particular length. More attention can be paid to this in future studies. Future studies should focus on one or two elements, such as communications and security, within the IWMCPS structure. These investigations can also be prolonged by improving the testbed established under the eight-year government initiative programs. Data from cyber-physical systems may be skewed if they were physically tampered with. Due to inadequate physical protection, the exposure of ICS components is categorized as a vulnerability that needs to be addressed in the future.

## Figures and Tables

**Figure 1 healthcare-11-00384-f001:**
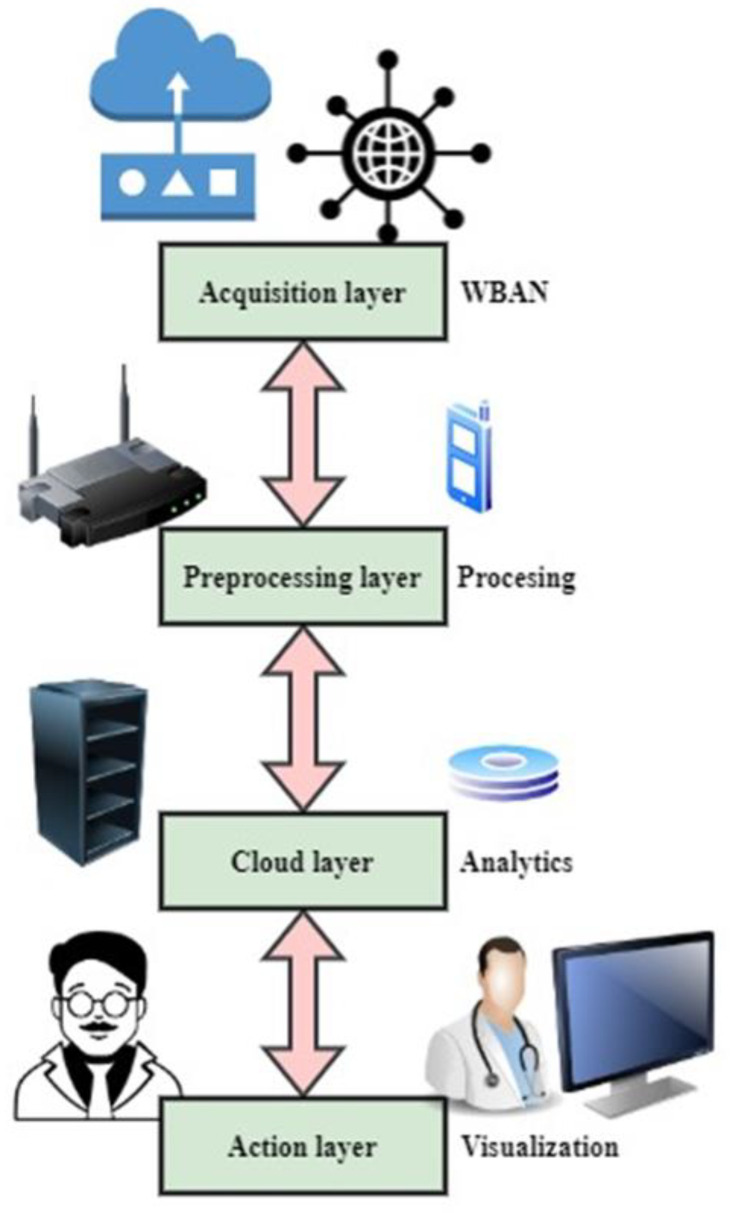
The proposed IWMCPS model architecture.

**Figure 2 healthcare-11-00384-f002:**
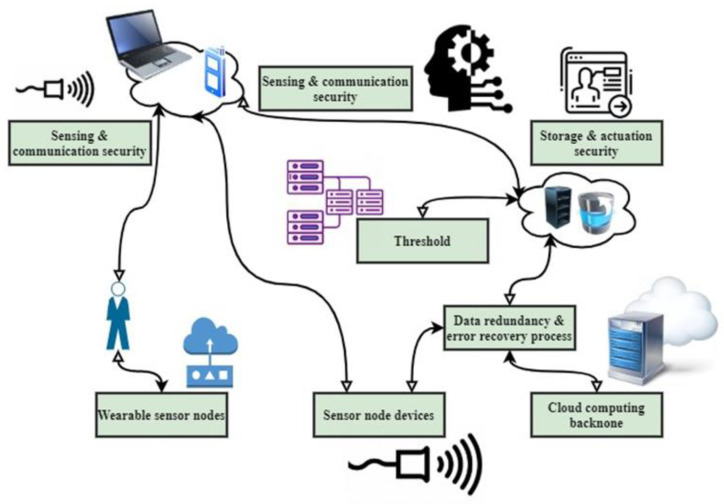
The workflow of the proposed IWMCPS model.

**Figure 3 healthcare-11-00384-f003:**
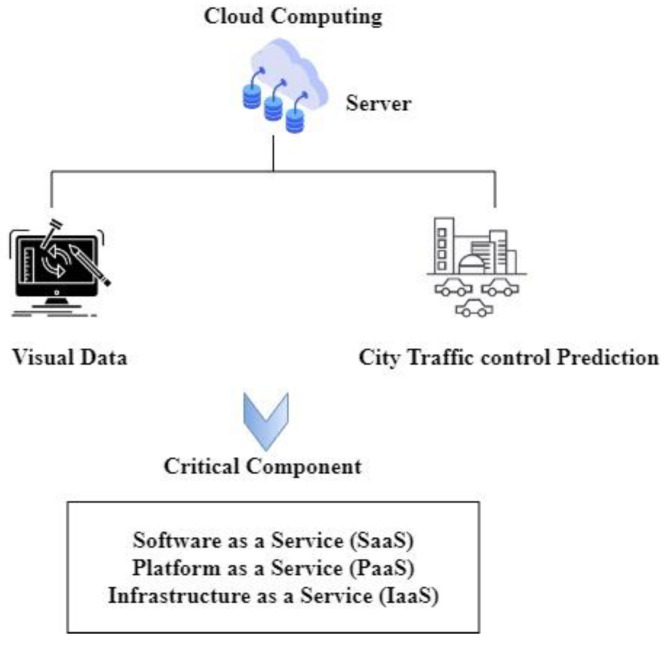
Critical components of the cloud computing architecture.

**Figure 4 healthcare-11-00384-f004:**
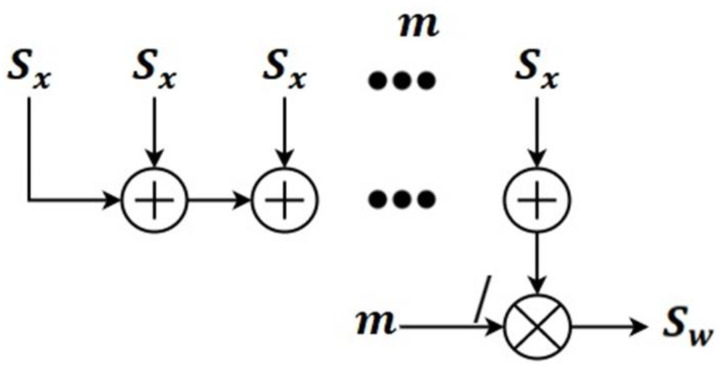
Pictorial representation of $S_{w}$.

**Figure 5 healthcare-11-00384-f005:**
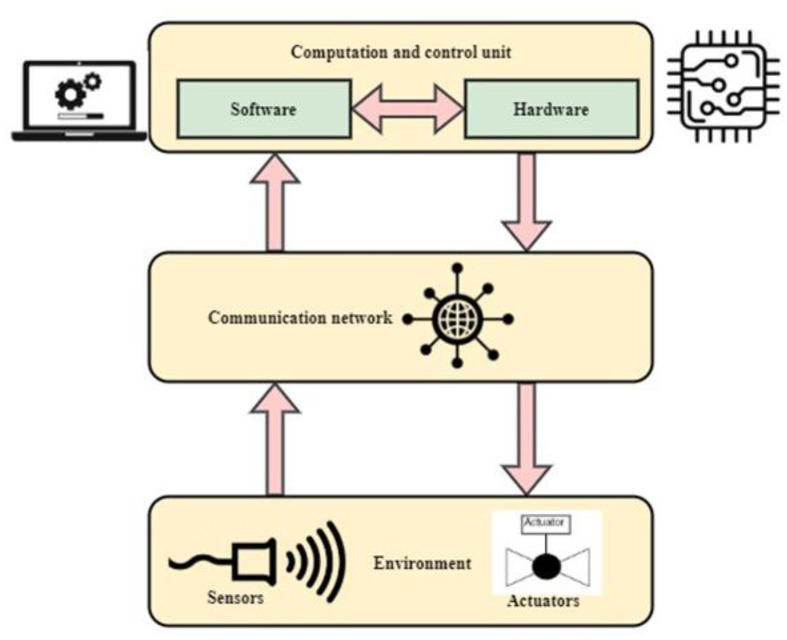
The data processing module of the proposed IWMCPS model.

**Figure 6 healthcare-11-00384-f006:**
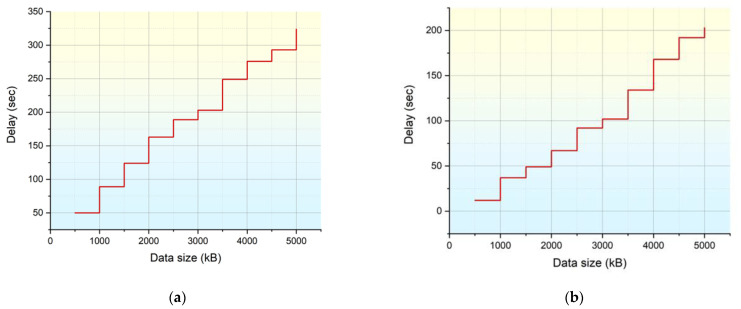
(**a**) Delay analysis of the existing CPS model and (**b**) delay analysis of the proposed IWMCPS model.

**Figure 7 healthcare-11-00384-f007:**
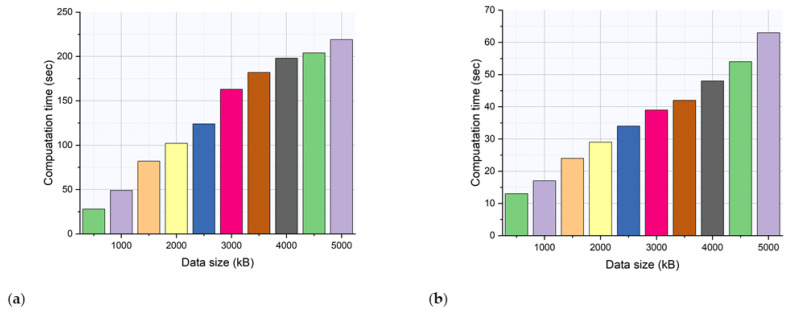
(**a**) Computation time analysis of the existing CPS model and (**b**) computation time analysis of the proposed IWMCPS model.

**Figure 8 healthcare-11-00384-f008:**
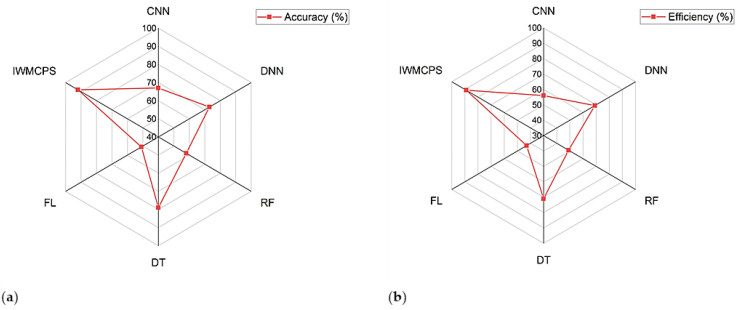
(**a**) Accuracy analysis and (**b**) efficiency analysis.

**Figure 9 healthcare-11-00384-f009:**
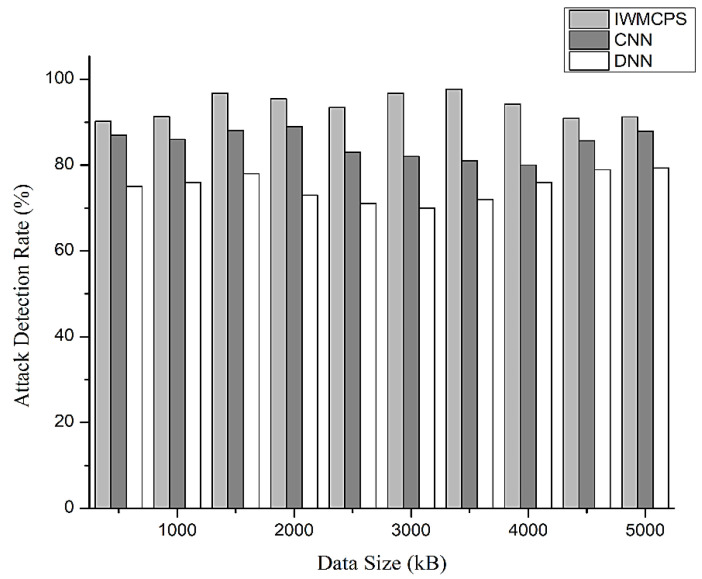
Attack detection rate of IWMCPS.

**Table 1 healthcare-11-00384-t001:** Simulation outcome analysis.

Method	Accuracy (%)	Efficiency (%)
CNN	67	56
DNN	73	69
RF	58	49
DT	79	71
FL	51	43
IWMCPS	92	89

**Table 2 healthcare-11-00384-t002:** Computation time analysis.

Data size (kB)	CPS (s)	IWMCPS (s)
500	28	13
1000	49	17
1500	82	24
2000	102	29
2500	124	34
3000	163	39
3500	182	42
4000	198	48
4500	204	54
5000	219	63

## Data Availability

Not applicable.

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
