# Peer review of "Improved Wireless Medical Cyber-Physical System (IWMCPS) Based on Machine Learning"

_healthcare, 2023, doi:10.3390/healthcare11030384_

Round 1
Reviewer 1 Report (New Reviewer)
The paper proposes a novel, Improved Wireless Medical Cyber-Physical System (IWMCPS) based on machine learning techniques, specifically deep neural networks, to detect potential security attacks on patient data collected using various methods. Initial findings suggest that the proposed IWMCPS system has the potential to detect security threats and potential attacks with high accuracy at low computational costs. I have a few concerns that I would like the authors to address to improve the quality of the manuscript.
Abstract
-
Throughout the abstract, there are minor issues with wording and sentence formation. For example, “MCPS recommend…” line 10 here, the word “recommend” does not fit the context of the sentence. I am guessing the authors wanted to use “represents.”
-
What do authors mean by “large attack surfaces?” higher likelihood for attack?
-
Why is “Denial of Service” in Upper Case while the other two are not (data modification.)
-
I am not sure about the importance of the following sentence. “The data used is well protected in the form of security aspects.”
-
I would highly recommend that the authors talk add some more details regarding the methods and conclusions to the abstract. It’s background heavy as of now.
General comments
-
The core work is interesting, but the paper is poorly written and lacks coherence. I would highly recommend authors take a step back and reorganize the Introduction to CPS section and have a technical editor review it.
-
Same comment as above for Background on Medical CPS. The literature review is robust, but the way the sentences are framed lacks coherence and does not align with the academic writing style.
-
Section 3 header, there is no space between “Proposed and Improved….”
-
“There are four standard MCPS architectures” It should be “four standard components to MCPS..”
-
Line 260 it should be “that enables….”
-
Line 287 it should be “stores and analyses….”
-
There are grammatical issues throughout the paper, so I am not going to list them all. Please proofread and have a technical editor review the paper.
-
Section 3.6 is well documented and discusses the important aspects. However, in Section 4, software analysis, I cannot find any details regarding the machine learning models. Model parameters and hyperparameters are not discussed at all.
-
Outputs in Tables and Figures suggest that IWMCPS is the best-performing case, but because of the lack of details on other models raises concerns. How did accuracy jump from 73% for DNN to 92% etc., is not discussed at all? Please add these details and discuss the findings in depth.
-
The authors do not discuss any limitations of the current and only briefly discuss the future steps.
Author Response
Response to 1nd Reviewer comments
We appreciate the time and effort that the reviewers have dedicated to providing their valuable feedback on our manuscript. We are grateful to the reviewers for their insightful comments on our paper. We have been able to incorporate changes to reflect almost all the suggestions provided by the reviewers.
We accepted all the suggestions, and we made the changes within the manuscript as per the reviewers’ comments. Here is a point-by-point response to the reviewers’ comments and concerns.
Comments and Suggestions for Authors
The paper proposes a novel, Improved Wireless Medical Cyber-Physical System (IWMCPS) based on machine learning techniques, specifically deep neural networks, to detect potential security attacks on patient data collected using various methods. Initial findings suggest that the proposed IWMCPS system has the potential to detect security threats and potential attacks with high accuracy at low computational costs. I have a few concerns that I would like the authors to address to improve the quality of the manuscript.
Abstract
- Throughout the abstract, there are minor issues with wording and sentence formation. For example, in “MCPS recommend…” line 10 here, the word “recommend” does not fit the context of the sentence. I am guessing the authors wanted to use “represents.”
Ans: Changed as represent
Medical Cyber-Physical Systems (MCPS) represent a platform in which patient health data is acquired by the emergent Internet of Things (IoT) sensors, preprocessed locally, and managed through improved machine intelligence algorithms.
- What do authors mean by “large attack surfaces?” higher likelihood for attack?
Ans:
Large attack surface discusses the All endpoint devices—from desktops to mobile phones to USB ports to carelessly disposed hard drives—are part of the cyber physical attack surface.
In Large attack surface the heterogeneity of devices included in these systems (such as mobile devices and body sensor nodes) is prone to many attacks.
- Why is “Denial of Service” in Upper Case while the other two are not (data modification.)
Ans:
The simplest and most damaging form of network attack is a denial of service (DoS). It has been utilized for Performing security control requires authorising the system to ignore standard requests to send and receive data using the communication channel's network resources.
This study evaluates our design with actual patient data against security attacks such as data modification, Denial of Service (DoS), and data injection
- I am not sure about the importance of the following sentence. “The data used is well protected in security aspects.”
Ans: The above-said sentence has been changed
The patient health data used in WMCPS must be well protected in the form of security aspects to overcome cyber-physical threats.
- I would highly recommend that the authors talk add some more details regarding the methods and conclusions to the abstract. It’s background heavy as of now.
Ans:
The three core elements in the proposed IWMCPS are the communication and monitoring core, the computational and safety core, and the real-time planning and administration of resources.
Experimental findings show that our model attains high detection accuracy of 92% and a lower computational time of 13secs with fewer error analyzes.
Patients' personal information must be shielded against intrusions to preserve their privacy and confidentiality. Therefore, every bit of information kept in the database needs to be safe from intrusion attempts. The proposed IWMCPS takes into account all relevant security concerns. This research, summarised the preceding three years of fieldwork by presenting an IWMCPS framework consisting of several components and subsystems. The IWMCPS architecture has been developed, as evidenced by a scenario including applications in the medical sector.
General comments
- The core work is interesting, but the paper is poorly written and lacks coherence. I would highly recommend authors take a step back and reorganize the Introduction to CPS section and have a technical editor review it.
Ans:
The potential for CPSs to positively impact society, the economy, the environment, and individual citizens have resulted in a surge of interest in the field in recent years. But, more importantly, the dominance and innovation in the information transmission systems area have emerged from the rapid advancements in computer, communication, and storage systems.
In addition, it is crucial to comprehend the external-process channels, such as the responses of human operators to the system, which includes the MCPS, medical sensors, and devices. High-confidence CPS and MCPS take into account the rising infrastructure problem to provide security and privacy. However, conventional methods of analysis are inadequate for either understanding their complexity or forecasting how the system would behave. For instance, as more and more Internet of Things (IoT) medical equipment and sensors interact with the real world and people, the bar for security, reliability, privacy, and safety increases dramatically for MCPS. Therefore, cutting-edge technology is necessary for engineering system designers to construct and hypothesise complex interdependencies. Where typical real-time performance is insufficient for geographically and big CPSs, novel connections between physical components and the cyber need new architectural paradigms. The availability of communication channels, which need security, is typically not taken into account in the CPS or the MCPS since the feedback and input are from the physical world. In terms of the CPS, this is a very important and distinguishing feature.
- Same comment as above for Background on Medical CPS. The literature review is robust, but the way the sentences are framed lacks coherence and does not align with the academic writing style.
Ans: Background based on Medical CPS is added
Maryam Bagheri et al. [43] focus on building trust in Machine Learning-based forecasts and creating a safety guarantee claim for Machine Learning controls in learning-enabled MCPS. Finally, as an example of a learning-enabled MCPS implementation, the safety promotion case is discussed for Artificial Pancreas Systems (APS) and offers a deep neural network implementation for predicting insulin levels.
All through the document it has been updated and highlighted.
As per the suggestion, the literature content has been optimized.
- Section 3 header, there is no space between “Proposed and Improved….”
Ans: Changed
Proposed Improved Wireless Medical Cyber-Physical System (IWMCPS)
- “There are four standard MCPS architectures” It should be “four standard components to MCPS..”
Ans: Changed
There are four standard components in MCPS architectures
- Line 260 should be “that enables….”
Ans:
Thus, the primary need is to enable a restricted power supply for Wireless Sensor Networks (WSN) to be maintained for a longer lifespan.
I has been updated as per the suggestion
- Line 287 should be “stores and analyses….”
Ans: As per the comments it has been updates as “The cloud layer stores and analyses the processed data”
- There are grammatical issues throughout the paper, so I am not going to list them all. Please proofread and have a technical editor review the paper.
Ans: Proof reading has been done
- Section 3.6 is well documented and discusses the important aspects. However, in Section 4, software analysis, I cannot find any details regarding the machine learning models. For example, model parameters and hyperparameters are not discussed at all.
Ans: added in the software analysis
Since the Model parameters are analysed based on the internal structure as well as validated based on the data automatically, The proposed method is compared with the machine learning techniques like CNN and DNN. Here, As a result, the overall performance in the form of parameters like attack detection rate is 98.92%, computation time is 13sec, Accuracy is 92%, and Efficiency is 89% for the proposed method. Further, whereas Hyperparameters are set manually in which The lambda architecture is implemented in this system, which is based on Apache Kafka and Apache Spark tools used for the simulation purpose.
- Outputs in Tables and Figures suggest that IWMCPS is the best-performing case, but because of the lack of details on other models raises concerns. How did accuracy jump from 73% for DNN to 92% etc., is not discussed at all? Please add these details and discuss the findings in depth.
Ans:
The Accuracy and the Efficiency of proposed IWMCPS have more Accuracy and Efficiency when compared to CNN, DNN, RF, DT, and FL. In addition, for particular data sizes, the Accuracy and Efficiency of the proposed method are higher. The proposed IWMCPS model with theoretical error calculation and avoidance model produces faster accuracy results
- The authors do not discuss any limitations of the current and only briefly discuss the future steps.
Ans: Added in conclusion
However, the data size is limited to a particular length. Therefore, more concentration can be given in future studies. Data from cyber-physical systems may be skewed if they were physically tampered with. In actuality, cyber-influenced physical assaults. Due to inadequate physical protection, the exposure of ICS components is categorised as a vulnerability which needs to be addressed in future.

Reviewer 2 Report (New Reviewer)
The manuscript proposes an Improved Wireless Medical Cyber-Physical System (IWMCPS) based on machine learning techniques. The heterogeneity of devices included in these systems (such as mobile devices and body sensor nodes) introduces large attack surfaces. However, several concerns, and comments are requested to revise them properly. These concerns are as follows.
-In the background section, authors are recommended to separate into several subsections in terms of the system model, security model, mathematical used, etc.
-In the introduction section, authors are requested to review these solutions based on machine learning and wireless communications. (i) DDoS Attacks Detection Using Machine Learning and Deep Learning Techniques: Analysis and Comparison; (ii) COVID-19 Vehicle Based an Efficient Mutual Authentication Scheme for 5G-Enabled Vehicular Fog Computing; (iii) MSR-DoS: Modular Square Root-based Scheme to Resist Denial of Service (DoS) Attacks in 5G-enabled Vehicular Networks
-The literature review section is missing. Thus, the authors are recommended to highlight the limitation and straights of related modern studies.
-Why use machine learning instead of deep learning in the manuscript?
-In order to prove your claim of contribution, the results should be analyzed and compared with recent works
-Provide figures/diagrams to show your proposal for clarification.
Author Response
Comments and Suggestions for Authors
The manuscript proposes an Improved Wireless Medical Cyber-Physical System (IWMCPS) based on machine learning techniques. The heterogeneity of devices included in these systems (such as mobile devices and body sensor nodes) introduces large attack surfaces. However, several concerns and comments are requested to revise them properly. These concerns are as follows.
-In the background section, authors are recommended to separate into several subsections in terms of the system model, security model, mathematical used, etc.
Ans: Subsection included in the background section
2.1. Cyber-Physical Systems
2.2. QoS problems in CPS
2.3. Medical Cyber-Physical Systems
-In the introduction section, authors are requested to review these solutions based on machine learning and wireless communications. (i) DDoS Attacks Detection Using Machine Learning and Deep Learning Techniques: Analysis and Comparison; (ii) COVID-19 Vehicle Based an Efficient Mutual Authentication Scheme for 5G-Enabled Vehicular Fog Computing; (iii) MSR-DoS: Modular Square Root-based Scheme to Resist Denial of Service (DoS) Attacks in 5G-enabled Vehicular Networks
Ans: Added
Distributed denial of service (DDoS) assaults pose a severe danger to the stability of the Internet. A Distributed Denial of Service (DDoS) attack aims to overwhelm a centralized system with so many queries or communications that it crashes, disrupting service for legitimate customers [15].
The advent of 5G has opened the way to non-contact automated healthcare surveillance, which could be facilitated by collecting and interpreting relevant vehicle locations using vehicular fog computing. This research presents a COVID-19 vehicle based on an effective joint identification system for 5G-enabled vehicle fog computing to contain the current automobile outbreak [16]
A modular square root-based defense mechanism is presented to prevent DoS attacks in 5G-enabled car networks. When applied to vehicle networks, the MSR-DoS solution guarantees source authenticity, message integrity, and pseudonym privacy cannot be linked, is traceable, and may be revoked. Under the tenets of burrows abadi Needham (BAN). The research and performance comparison reveal that the MSR-DoS system has lower communication and computational expenses than the latest current works [17].
-The literature review section is missing. Thus, the authors are recommended to highlight the limitation and straights of related modern studies.
Ans: Highlighted
Integrating medical care from many perspectives increases the confidentiality and privacy of patients' real-time databases. In addition, security and data quality must be considered during the configuration phase of any activity involving security measurements against different attacks. Security issues are the major drawback of CPS. IWMCPS addresses those security gaps through strategic Planning based on the cloud storage and safety components.
-Why use machine learning instead of deep learning in the manuscript?
Ans: The proposed method focus more on the machine learning techniques, for easy training of the model and in terms of accuracy measurement machine learning method is implemented
-In order to prove your claim of contribution, the results should be analyzed and compared with recent works
Ans: Included in the result section
The Accuracy and the Efficiency of proposed IWMCPS have more Accuracy and Efficiency when compared to CNN, DNN, RF, DT, and FL. In addition, for particular data sizes, the Accuracy and Efficiency of the proposed method are higher. Since the Model parameters are analysed based on the internal structure as well as validated based on the data automatically, The proposed method is compared with the machine learning techniques like CNN and DNN. Further, whereas Hyperparameters are set manually in which The lambda architecture is implemented in this system, which is based on Apache Kafka and Apache Spark tools used for the simulation purpose.
-Provide figures/diagrams to show your proposal for clarification.
Ans: The proposed method is clearly explained in figure 1
Round 2
Reviewer 1 Report (New Reviewer)
Thanks for addressing all my concerns. The quality of the manuscript is much better.
Reviewer 2 Report (New Reviewer)
done
This manuscript is a resubmission of an earlier submission. The following is a list of the peer review reports and author responses from that submission.
Round 1
Reviewer 1 Report
Dear Authors,
Please address the following concerns.
1- You should clearly declare the major innovations of your study compared with other state-of-the-art studies as well as the numerical values should be mentioned in the abstract section.
2- some grammatical and layout errors occurred in the paper which should be corrected.
3- In the introduction section, you should mention the importance of the application of Machine Learning in the other field of engineering. Please include some new research papers from various engineering fields in this regard. (e.g. 1- Savari, M. Amin, and Hadi Jahanirad. "NN-SSTA: A deep neural network approach for statistical static timing analysis." Expert Systems with Applications 149 (2020): 113309. 2-Rahimi, H., and Hadi Jahanirad. "An evolutionary approach to implement logic circuits on three dimensional FPGAs." Expert Systems with Applications 174 (2021): 114780. 3-Vaziri, Mobin, Mohammad Mehdi Rahimifar, and Hadi Jahanirad. "An Enhanced Chaotic System Based Color Image Encryption using DNA Encoding." In 2022 30th International Conference on Electrical Engineering (ICEE), pp. 128-133. IEEE, 2022.)
Author Response
- You should declare the major innovations of your study compared with other state-of-the-art studies as well as the numerical values should be mentioned in the abstract section.
Ans: a major drawback of the existing method is included in the abstract
Cyber-Physical Systems are essential to the healthcare sector, vulnerable to information thefts and cyber-attacks because they are life-critical and context-aware. Reliability, confidence, security, and transparency are some of the issues that must be addressed in the growing field of MCPS study. To overcome the problems mentioned above, this research suggests an Improved Wireless Medical Cyber-Physical System (IWMCPS) based on machine learning techniques.
Experimental findings show that our model attains high detection accuracy of 92% and a lower computational time of 13secs.
- some grammatical and layout errors occurred in the paper which should be corrected.
Ans: Proof reading is done
- In the introduction section, you should mention the importance of the application of Machine Learning in the other field of engineering. Please include some new research papers from various engineering fields in this regard. (e.g. 1- Savari, M. Amin, and Hadi Jahanirad. "NN-SSTA: A deep neural network approach for statistical static timing analysis." Expert Systems with Applications149 (2020): 113309. 2-Rahimi, H., and Hadi Jahanirad. "An evolutionary approach to implement logic circuits on three dimensional FPGAs." Expert Systems with Applications 174 (2021): 114780. 3-Vaziri, Mobin, Mohammad Mehdi Rahimifar, and Hadi Jahanirad. "An Enhanced Chaotic System Based Color Image Encryption using DNA Encoding." In 2022 30th International Conference on Electrical Engineering (ICEE), pp. 128-133. IEEE, 2022.)
Ans: the importance of the application of Machine Learning is included in the introduction
Automated malware recognition and prevention are crucial in healthcare cyber-physical methods for detecting and categorizing malicious threats. Machine learning techniques are used for detecting and classifying the different attacks for data theft from the medical record. The data stored in the cloud should be confidential for future use. Machine learning techniques in the medical field are a powerful tool for detecting and classifying attacks.
0ne of the reference is included in the related work section
A new method of secure color image encryption using a chaotic system and decoding with a simple, one-round procedure is offered in [36]. In the first step, a hash function creates a cryptographic key that would be utilized in both the confusion and distribution phases. The experimental analysis is more secure than previous approaches and is immune to various attacks, differential attacks, and occlusion threats.
Reviewer 2 Report
1. Abstract needs revision. Novelty is to be stated clearly. Results to be provided in terms of metrics. Number of samples tested etc must be mentioned. Its too generic
2. Atleast 5 index terms needed.
3. Intro section must contain back ground, motivation, novel features of the work etc.
4. Related works: Inferences from the previous studies must be spelled out and how this work addresses those gaps can be included.
5. Comparison : How the control parameters are set for CNN, DNN etc? How the samples are divided and tested?
Author Response
- Abstract needs revision. Novelty is to be stated clearly. Results to be provided in terms of metrics. Number of samples tested etc must be mentioned. Its too generic
Ans: Included in the abstract
The IWMCPS method is based on a patient-centric architecture that preserves the device known as the end-users smartphones and end-user control data exchange accessibility. The data used is well protected in the form of security aspects.
The sample of data size used is mentioned in the result and discussion section
- Atleast 5 index terms needed.
Ans: Index term added
Security Schemes; Machine learning; Medical Cyber-Physical Systems, attacks, data, classification
- Intro section must contain back ground, motivation, novel features of the work etc.
Ans: Inlcuded in the intro section
To address the threats and ensure the safety of patient data, CPS with machine learning techniques is implemented for monitoring healthcare networking with anomalies and malicious activity. Malware, encryption, data theft, hacking, and threat actors remain common in the healthcare industry.
Already the novel features of the work is mentioned as the contribution of the proposed work
- Related works: Inferences from the previous studies must be spelled out and how this work addresses those gaps can be included.
Ans: included in the related works
Integrating medical care from many perspectives increases the confidentiality and privacy of patients' real-time databases. In addition, security and data quality must be considered during the configuration phase of any activity involving security measurements against different attacks. Security issues are the major drawback of CPS. IWMCPS addresses those security gaps through strategic Planning based on the cloud storage and safety components.
- Comparison : How the control parameters are set for CNN, DNN etc? How are the samples divided and tested?
Ans: Included in the result section
The proposed model is evaluated in terms of accuracy, efficiency, delay, and time consumption. The data size variations are mentioned in the delay graph.
The proposed IWMCPS model is compared with CNN and DNN based on classification accuracy and efficiency.
Reviewer 3 Report
This research suggests an Improved Wireless Medical Cyber-Physical System (IWMCPS) based on machine learning techniques.
The study is interesting and the selected papers are good.
However, the paper needs to acquire more quality in terms of cited papers.
For this purpose I suggest to include the following paper among the cited papers because, in my opinion, it is crucial introducing some of formal methods for wireless sensor networks (WSNs) related to machine learning approaches:
"Heuristic strategies for assessing wireless sensor network resiliency: an event-based formal approach". .Journal of Heuristics 21 (2), 145-175
I am confident if the authors add the citation and they motivate the importance to consider it, the paper will acquire more quality for the publication.
Author Response
The study is interesting and the selected papers are good.
Thank you for the positive comments
However, the paper needs to acquire more quality in terms of cited papers.
For this purpose I suggest to include the following paper among the cited papers because, in my opinion, it is crucial introducing some of formal methods for wireless sensor networks (WSNs) related to machine learning approaches:
"Heuristic strategies for assessing wireless sensor network resiliency: an event-based formal approach". .Journal of Heuristics 21 (2), 145-175
Ans: Included in the related work section
Two practical techniques, what-if analysis, and robustness assessment, are offered in [37]. People enable to lead designers for ideal Wireless Sensor Network implementation options from the standpoint of connectivity and information provision resilience, employing a formalized method centered on the incident calculus formal languages.
I am confident if the authors add the citation and they motivate the importance to consider it, the paper will acquire more quality for the publication.
Thankyou for the positive comments